| 1  | Electrical Resistivity Tomography surveys for the geoelectric                                                                                                               |
|----|-----------------------------------------------------------------------------------------------------------------------------------------------------------------------------|
| 2  | characterization of the Montaguto landslide (southern Italy)                                                                                                                |
| 3  |                                                                                                                                                                             |
| 4  | * Jessica Bellanova <sup>1</sup> , Giuseppe Calamita <sup>1</sup> , Alessandro Giocoli <sup>2</sup> , Raffaele Luongo <sup>1</sup> , Angela Perrone <sup>1</sup> , Vincenzo |
| 5  | Lapenna <sup>1</sup> , Sabatino Piscitelli <sup>1</sup>                                                                                                                     |
| 6  | <sup>1</sup> Institute of Methodologies for Environmental Analysis (IMAA-CNR), Tito Scalo (PZ), Italy.                                                                      |
| 7  | <sup>2</sup> Italian National Agency for New Technologies, Energy and Sustainable Economic Development (ENEA), Rotondella                                                   |
| 8  | (MT), Italy                                                                                                                                                                 |
| 9  |                                                                                                                                                                             |
| 10 |                                                                                                                                                                             |
| 11 | *Corresponding author                                                                                                                                                       |
| 12 | Bellanova Jessica                                                                                                                                                           |
| 13 | Institute of Methodologies for Environmental Analysis                                                                                                                       |
| 14 | Italian National Research Council                                                                                                                                           |
| 15 | C.da S. Loja                                                                                                                                                                |
| 16 | 85050, Tito (PZ)                                                                                                                                                            |
| 17 | Italy                                                                                                                                                                       |
| 18 | jessica.bellanova@imaa.cnr.it                                                                                                                                               |
| 19 |                                                                                                                                                                             |
| 20 |                                                                                                                                                                             |

# 21 Abstract

22 This paper reports the results of a geoelectrical survey carried out to investigate the Montaguto 23 earth-flow, located in the southern Apennines (Campania Region, southern Italy). The aim of the 24 survey was to reconstruct the geometry of the landslide body, to improve the knowledge on the 25 geological setting and to indirectly test the effectiveness of a drainage system. Although electrical 26 resistivity contrasts in the electrical images were not very pronounced, due to the lithological 27 characteristic of the outcropping lithotypes, it was possible to observe the presence of both lateral 28 and vertical discontinuities that were associated with lithological boundaries, physical variation of 29 the same material and sliding surfaces. The geoelectrical information obtained was provided to the 30 Italian National Civil Protection Department technicians and was considered for the planning of 31 more appropriate actions for the stabilization and safety of the slide. 32 33 Keywords: Electrical Resistivity Tomography; Landslide; Earth-flow; Montaguto; southern Italy

34

Natural Hazards and Earth System Sciences Discussions

#### 36 1. Introduction

- The Montaguto landslide, located in southern Apennines (Campania Region, southern Italy), is one of the larger and complex earth-flow in Europe (Fig. 1). It was active for almost 60 years starting from, at least, 1954. Long periods of relatively slow movement and shorter periods of relatively rapid movement periodically have followed one another in the earth-flow activity (Guerriero et al., 2013).
- 42
- 43

Figure 1 - Location of the Montaguto earth-flow, southern Apennines (Campania Region, southern Italy). White line: landslide boundary. Black line: railway. Yellow line: road SS90. Blue line: Cervaro River.

45

During the mid spring season of 2006, the most extensive reported slope failure started; an estimated volume of  $6 \times 10^6$  m<sup>3</sup> earth-flow was activated. Four years later, in the spring of 2010, the earth-flow reached the Cervaro River valley, obstructing and strongly damaging the strategic National Railway infrastructure, connecting the towns of Naples and Bari, and the SS90 National Road, connecting Campania and Apulia Regions (Ventura et al., 2011; Guerriero et al., 2013).

Considerable efforts were carried by the Italian National Civil Protection Department (DPC) to

tackle the emergency. Actions like artificial drainages, removal of slide material from the toe, etc.,

have been taking place since then, in order to mitigate the effects of the mass movement.

Notwithstanding the resulting slowdown of the earth-flow obtained, further coordinated actions are

yet ongoing to ensure safer conditions to the railway and road infrastructures. However, in order to

implement a well structured and comprehensive plan of intervention actions, further relevant

geological, geotechnical and geophysical details (mechanical characteristics of the material,

geometry of the body, etc.) are needed.

This paper reports the results of two geoelectrical surveys carried out in the area, in July 2011 and 60 October 2012. As explicitly required by the DPC, the first survey was focused on the upper portion 61 of the landslide body, to check a drainage intervention and to obtain the preliminary geophysical 62 information on the terrains involved in the movement. The second survey was carried out in the 63 central part of the landslide, between about 600 and 520 m a.s.l. that, despite the drainage 64 interventions, was characterized by a trend of continuous movement. This state of activity made it different from other sectors of the landslide body and needed a deepening of the monitoring 65 66 activities (Lollino et al., 2013; Lollino et al., 2014).

#### 68 2. Geological and Geomorphological setting

- The area affected by the Montaguto earth-flow is located in a region known as "Daunia Apennines"
- in the eastern part of southern Apennines.
- The Daunia Apennines belong to the highly deformed transition area between the frontal thrusts of

the Apennine chain and the western part of the foredeep (Crostella and Vezzani, 1964; Dazzaro et

- al., 1988). The lithological units, present in this area, are characterized by the presence of flysch
- units of Miocene age, rich in clay component, intensely deformed, as a result of the tectonic history
  of the Apennines (Amore et al., 1998; Di Nocera et al., 2011), and prone to landsliding. Usually, the
- activity of landslides is characterized by seasonal remobilizations of slope movements, typicallydue to rainfall events.
- In the study area crop out the Faeto Flysch (FF), belonging to the Daunia Unit (Crostella and Vezzani, 1964), the Villamaina Unit (FV) (Di Nocera and Torre, 1987; Pescatore et al., 1996), colluvial deposit (d) and alluvial sediments (a). The Faeto Flysch and the unconformable overlying
- Villamaina Unit crop out in the upper part and in the middle-lower sector of the landslide,
- respectively; the alluvial sediments are present in the Cervaro River valley (Guerriero et al., 2014)
- (Fig. 2).

The Faeto Flysch, aged from Langhian to Tortonian, is composed by basinal and shelf margin facies 85 and consists of three lithofacies, which from the bottom upward are: a calcareous-clayey-marly 86 succession (FFa), composed by calcarenite and clay, passing upward to calcarenite, calcirudite and 87 white marl; a calcareous-marly succession (FFb), represented by a dense alternation of calcarenite 88 and marl, and a clayey-marly-calcareous succession (FFc), that consist of calcarenite, white marl 89 and green clay (Santo and Senatore, 1988). The slope affected by the study earth-flow is only 90 characterized by the outcropping of the basal member of the Faeto Flysch (FFa) (aged Burdigalian 91 sup. - Langhian inf.), which has, locally, a prevalently calcareous-marly (FFa1) or clayey (FFa2) 92 composition.

The Villamaina Unit, Early Messinian in age, is made up of conglomerates (FVa), sandstones not
very well cemented with a few clay beds (FVb) and, upward, brownish-gray sandy with silty clay
beds (FVc) (Lollino et al., 2014).

The recent 2010 Montaguto landslide is characterized by a length of  $3.1 \times 10^3$  m, a width ranging between 45 and 420 m and an aerial extension of about  $6.6 \times 10^5$  m<sup>2</sup> (~66 ha). It was estimated a volume of displaced material of about  $4 \times 10^6$  m<sup>3</sup> and a sliding surface depth varying from about 5 m, near the channel area, to 20-30 m, at the toe (Ventura et al., 2011; Giordan et al., 2013; Guerriero et al., 2013; Lollino et al., 2014). As stated by Ventura et al. (2011), the depth of the water table roughly corresponds to the thickness of sliding material with sag ponds occurring in the Nat. Hazards Earth Syst. Sci. Discuss., doi:10.5194/nhess-2016-28, 2016 Manuscript under review for journal Nat. Hazards Earth Syst. Sci. Published: 29 February 2016

© Author(s) 2016. CC-BY 3.0 License.

- upper and central zone. The altitude gap between the landslide head scarp, 830 m a.s.l., and the toe,
- 420 a.s.l., is about 410 m (Giordan et al., 2013).
- The reported velocities of most movement, from 1954 to 2010, ranged from 1 - 2 mm/month to 2 - 2
- 5 cm/day. A sharp increase was registered during the large mobilization on both 2006 and 2010,
- from 1 m/day to 1 m/hour, as reported by Guerriero et al. (2013), or 5 m/day, as reported by
- Giordan et al. (2013).

Figure 2 - Geological map of the Montaguto earth-flow. Legend: colluvial deposits (d); alluvial deposits (a); Villamaina Unit (FVa: conglomerate; FVb: sandstone and clay; FVc: sand and silty clay); Faeto Flysch (FFa: calcarenite, clay and marl); line with hachures: normal fault (dashed when inferred); line with triangles: axis of fold structure. The white area indicates the active earth-flow. The pink area indicates the inactive toe of the old landslide (IT). Blue lines: profiles of the ERT carried out in July 2011. Red lines: profiles of ERT carried out in October 2012. Green dot: borehole. Blue triangle: piezometers. Coordinates in UTM 33 N are shown (modified from Guerriero et al., 2014).

#### 112 3. The Electrical Resistivity Tomography method

Electrical Resistivity Tomography (ERT) technique has been largely applied for the investigation of

landslide areas (McCann and Foster, 1990; Gallipoli et al., 2000; Hack 2000; Lapenna et al., 2003;

Perrone et al., 2004; Lapenna et al., 2005; Lebourg et al., 2005; Perrone et al., 2006; Naudet et al.,

2008; Chambers et al., 2011; Perrone et al., 2014), providing useful information on the geometrical

characteristics of the investigated body and on potentially instable areas, due to the high water 118 content.

Resistivity measurements are made by injecting a controlled current into the ground through two

steel electrodes and measuring the potential drop at other two electrodes. An apparent resistivity

value ( $\rho_a$ ) is calculated taking into account the intensity of the injected current (I), the potential drop

(V) and a geometric coefficient (k) related to the spatial electrode configuration,  $\rho_a = k \cdot V/I$ . Different

electrode arrays, such as Wenner, Schlumberger, dipole-dipole, etc., can be used for ERT surveys.

To obtain a subsurface image of the electrical resistivity, the apparent electrical resistivity data have

to be inverted in true electrical resistivity values by means of specific inversion software.

In this work, apparent electrical resistivity data were acquired through a multi-electrode system (48

electrodes) using a Syscal Junior (Iris Instruments) resistivity meter connected to a multicore cable.

A constant spacing (a) of 5 m between adjacent electrodes was used and a Wenner-Schlumberger

- (WS) array was adopted with different combinations of dipole length (1a, 2a and 3a) and number of
- depth levels "n" (n  $\leq$  6). The investigation depths were about 40 m. Data noise was assessed by
- means of repeatability tests (Robert et al., 2011). Five to ten stacked measurements were carried out

for each point and the respective relative standard deviation (Dev parameter) was estimated. The 133 resistivity values characterized by a Dev parameter greater than 1% and all of the obvious outliers 134 were removed. The apparent electrical resistivity data were inverted using the RES2DINV software 135 (Loke, 2001) to obtain the 2D electrical resistivity images of the subsurface. The inversion routine is based on the smoothness-constrained least-squares inversion method implemented by using a 136 137 quasi-Newton optimisation technique (Sasaki, 1992; Loke and Barker, 1996). The optimisation 138 method adjusts the 2D electrical resistivity model trying to iteratively reduce the difference between 139 the calculated and measured apparent resistivity values. The root-mean-squared (RMS) error 140 provides a measurement of this difference. 141 All the ERT profiles, each with a length of 235 m, were placed perpendicularly to the main axis of

the channel area of the landslide (Fig. 2). In particular, in the first field survey, on July 2011, three ERT were carried out in the upper-zone of the channel area between 700 m and 620 m a.s.l. The main aim of this survey was to check the functionality of a drainage trench located in the area and to obtain preliminary information on the geoelectrical characteristics of the material involved in the

146 movement.

More than one year later, on October 2012, eleven ERT were realized in the central part of channel area, along parallel profiles spaced 50-60 m apart (Fig. 2). The aim of this survey was to characterize the geometry of this portion of the landslide, to improve the knowledge about the geological setting and to indirectly test the effectiveness of the specifically installed drainage system. This latter represented a very important information for the technicians of DPC, because this portion of landslide, despite the drainage interventions carried out, is characterized by a trend of continuous movement (Lollino et al., 2013; Lollino et al., 2014).

### 155 **4. Results**

Here the results obtained during the two surveys carried out on July 2011 and October 2012 arediscussed.

For all the ERT, the range of the electrical resistivity values is guite limited, varying between 3 and 159 more than 34  $\Omega$ m. Generally, since the electrical resistivity of a rock is controlled by different 160 factors (water content, porosity, clay content, etc.), there are wide ranges in electrical resistivity for 161 any particular rock type and, accordingly, electrical resistivity values cannot be directly interpreted 162 in terms of lithology. For these reasons, we used data from literature (Giocoli et al., 2008; 163 Mucciarelli et al., 2009), geological surveys and exploratory boreholes to calibrate the ERT and to 164 directly correlate electrical resistivity values with the lithostratigraphic characteristics. Thus, the 165 following electrical resistivity ranges were assigned:  $\rho > 12 \Omega m$  to the FFa1,  $\rho

$> 20 \ \Omega m$  to FVb and  $\rho < 8 \ \Omega m$  to FVc. In particular, the active landslide material is characterized by electrical resistivity values ranging between 6 and 12  $\Omega$ m, whereas the inactive earth-flow toe 167 168 and of the old earth-flow show  $\rho > 8 \Omega m$  and  $\rho < 12 \Omega m$ , respectively.

July 2011: first survey

Figure 2 shows the profiles (blue lines) along that ERT were carried out during the first 172 measurement campaign on July 2011. The profiles cross (active and inactive) landslide material and 173 terrains belonging to the FF and FV. The lithological composition of these formations contributes to 174 justify the low resistivity range characterizing the ERT. 175 Despite low resistivity contrasts, the three ERT allowed us to define the geometry of active and

inactive landslide bodies, to identify sub-vertical discontinuities, often corresponding with the 177 lateral limits of the earth-flow, and to locate areas characterized by higher water content (Fig. 3).

Figure 3 - Resistivity models of the three ERT carried out across the Montaguto landslide in July 2011.

181

In particular, ERT 1 was placed parallel to one of the first drainage trenches, installed in the 183 investigated area at a quote of about 700 m a.s.l., and shows both vertical and horizontal resistivity 184 variations. In detail, between 85 and 180 m a relatively resistive superficial sector (8 <  $\rho$  < 25  $\Omega$ m), 185 about 10-12 m thick, likely due to the drainage trench and active landslide material, is clearly 186 identifiable. At the bottom, a relatively conductive layer ( $\rho < 6 \Omega m$ ), laterally limited by more 187 resistive zones ( $\rho > 12 \ \Omega m$ ), could be associated with the clayey lithofacies (FFa2). By comparing 188 the ERT with geological information, the more resistive zone located in SE portion could be related 189 to the calcareous-marly lithofacies (FFa1) and the sub-vertical resistivity discontinuity could be due 190 to the presence of a NE-SW normal fault, as reported in the map of figure 2, according to Guerriero 191 et al. (2014). In the NW part of the ERT, the deep more resistive zone can be associated with FFa1. 192 Finally, the shallow lenticular low resistivity zone in the NW sector can be interpreted as FFa2.

ERT 2 was carried out between 625-650 m a.s.l. It is characterized by two shallow areas of 194 conductive material ( $\rho < 12 \ \Omega m$ ) with lenticular shape, overlying a relatively resistive material ( $\rho > 12 \ \Omega m$ ) 195 12  $\Omega$ m). The first one, in the WSW sector of the ERT up to 100 m from the origin of the profile, 196 may be associated with the inactive landslide body (IT in Fig. 2). The second one, in the central 197 portion of the ERT between 105 m and 170 m from the origin and with a maximum thickness of 10 198 m, is related to the active landslide. The more resistive material, in the deep part of ERT, is

associated with FFa1. Finally, the conductive area located at the eastern part of the ERT andbounded by the NE-SW normal fault can be associated with FFa2.

ERT 3 was realized between 613 - 628 m a.s.l. and is characterized by a chaotic resistivity 202 distribution with weak lateral discontinuities. Between 85 m and 210 m from the origin of the 203 profile, a shallow (max 8 m thick) relative resistive material is associated with the active earth-flow 204 underlying a more conductive material, probably related to an old inactive landslide body. In the 205 WSW sector, the shallow moderately resistive material ( $\rho > 8 \Omega m$ ), with a maximum thickness of 206 about 12 m, is associated with the inactive earth-flow toe (IT) (Guerriero et al., 2014). The medium 207 resistive material, which characterizes the bottom and the ENE sector of the ERT, can be related to 208 FFa1.

#### 210 October 2012: second survey

During the second survey, eleven ERT were carried out, with direction transversal to the landslide body along profiles parallel to each other and spaced approximately 50-60 m, in the central part of the channel area (Fig. 2). Before the geophysical survey, several actions (excavation, surface drainage, etc) aimed at the stabilization of the landslide in this sector of slope were adopted However, despite the drainage interventions carried out, this sector is characterized by a trend of continuous movement (Lollino et al., 2013; Lollino et al., 2014). All the electrical images are reported in figures 4 and 5 and show almost the same resistivity

All the electrical images are reported in figures 4 and 5 and show almost the same resistivity 218 pattern: the central part is always characterized by conductive material of lenticular shape, confined 219 within more resistive material by means of sub-vertical contacts. Only ERT 11 shows a different 220 resistivity configuration, probably because performed entirely inside the landslide body.

222

Figure 4 - Resistivity models of six ERT carried out in the central part of the channel area of the Montaguto landslide in October 2012.

Figure 5 - Resistivity models of five ERT carried out in the central part of the channel area of the Montaguto landslide
 in October 2012

All the resistivity models well highlight the presence of drainage channels that show up as very shallow resistive nuclei. Shallow moderate resistive material ( $6 

ERT, can be associated with the presence of a higher water content than surrounding material. This assumption is also supported by the piezometric information coming from boreholes S8, S7 and S6 and piezometers P1 and P2 (Lollino et al., 2014). The lenticular shape of this material could be also related to an old inactive landslide body, below the currently active one, reaching a maximum thickness of about 30 m. This old landslide material seems to be confined in a paleo-channel characterized by relatively resistive boundaries. The more resistive material in the deep part of ERT could be related to FFa1 (ERT 1 to ERT 4) or to FVb (ERT 5 to ERT 9).

The NE part of almost all ERT is characterized by high electrical resistivity values that are associated with material not affected by the movement and belonging to FFa1 (ERT 1 to ERT 5) and to FVb (ERT 6 to ERT 10). Conductive material visible at the end of ERT 1 - ERT 4 profiles could be related to FFa2.

The sub-vertical resistivity discontinuities in the NE sector of all ERT (except for ERT 11) could be associated with the extension of the NE-SW normal fault, partially reported in figure 3 in Guerriero et al. (2014).

The SW portions of all ERT consist of low-medium resistive material related to the sandy with siltyclay beds (FVc).

## 249 **5.** Conclusions

This paper reports the results of two geoelectrical surveys carried out on the Montaguto landslide, in order to give a contribution in the geometrical characterization of the landslide body and in the definition of the geological setting. In addition, the effectiveness of the drainage system was indirectly tested.

Although electrical resistivity contrasts in the ERT images are not very pronounced, it was possible to observe the presence of both lateral and vertical discontinuities, which can be ascribed to lithological boundaries and/or physical variations of the same material with varying water content.

Regarding the geometrical characterization of landslide body and the reconstruction of geological setting in the channel area, the resistivity distribution in ERT images has highlighted the following points:

- the current active landslide material, reaching a maximum thickness of 15 m, is characterized by low-medium resistivity values ( $6 

- the lateral resistivity discontinuities, especially characterizing the NE sector of the ERT obtained
on October 2012, represent the lateral limits of the both active and old landslide body. In some
cases, these lateral limits are sub-vertical and can be associated with the presence of tectonic
structures (normal fault) according to the morphology of the slope and the previous geological
studies carried out in the area. Conversely, in the SW portion the superficial lateral limits of the
landslide body not seem to be marked by clear resistivity contrasts, due to the outcropping
lithotypes and to the presence of a high water content.

- ERT allowed the identification of the all drainage channels built in the upper and middle sector of landslide body. These structures are located in the first very shallow layers of the subsoil and are characterized by relatively high resistivity values ( $\rho > 12 \ \Omega m$ ). Considering that the material included between the drainage channels is characterized by medium resistivity values ( $6

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
