# Peer review of "(untitled)"

_Natural Hazards and Earth System Sciences, 2016_

## Referee Comment (RC1) · Anonymous Referee #1 · 16 Mar 2016

**Electrical Resistivity Tomography surveys for the geoelectric characterization of the Montaguto landslide (southern Italy)**

**Pag.3 line 49**

I think that translation of Puglia Region in "Apulia" Region is not completely correct when, in the same phrase, Campania is written in Italian language.

**Pag.3 Line52/53**

I think that for a completely explanation of the studies in the Montaguto Earthflow, is necessary to insert a citation of the complex monitoring activities according with (*Giordan et Al. 2013, Lollino et Al. 2014*)

**Pag.3 Line59/60**

According with the plane of the ERT surveys (fig.2) , I think that is more correct: "…was focused on the upper portion of *the landslide channel area*" because the upper part of the landslide body (from about 750 to 900 m asl) was not covered by ERT surveys

**Figure 3**

On the ERT 3 picture, we observe the S4 borehole but in the legend is not present. The Piezo 1 (P1) is instead present.

**Figure 4**

I think that 6 ERT sections in one A4 page are too many. I think that the correct number can be 3 or 4 (as in Figure 3).

**Figure 3,4**

To increase the comprehension of the drainage channel effects, the pictures are too small. My suggestion: create a zoomed box centered on the drainage channel.

The S4,S6,S7 piezometer explain the water table or pore water pressure? The difference between W.T. and P.W.P. is very important on this type of terrains

**Pag.9 - Conclusions - Line 251/252**

Regarding the effectiveness of the drainage system, I'm not completely sure that with this surveys is possible to make this important and critical information. I think that the your data explain the first effects of the drainage systems but just with many multi-temporal surveys is possible to explain the effectiveness of the complex drainage systems installed in this terrain. Therefore, my suggestion to modify the phrase.

---

## Referee Comment (RC2) · Anonymous Referee #2 · 24 Mar 2016

1. Lines 46-47: the sentence seems relatively questionable, since it is quite unrealistic that the slope failure started in the mid spring of 2006. Based on the historical data, the lower portion of the earth-flow mass obstructed the SS90 National Road on April 2006 and this means that the earthflow was activated some months before, with a run-out of several hundred of meters. In 2010 there was a strong reactivation of the landslide mass already accumulated at the toe of the slope in the previous years.

2. Line 97: a maximum value of earth-flow width equal to 420 m seems too high. Probably, the authors refer also to a portion of the landslide mass that invaded the valley located at the right-hand side of the proper valley occupied by the landslide mass, but that portion should not be considered as a real part of the main earth-flow

body. Therefore, a more precise value of the landslide width should be about 150-200 m.

3. Throughout the manuscript the authors distinguish between the active landslide material and the underlying inactive one, only based on the ERT data interpretation. This seems too strong, since there is not a clear instrumental evidence of what is the active part of the landslide. Moreover, for this type of landslides it is quite frequent that recently-activated landslide surges travelling above pre-existing landslide masses that rest along the landslide channel are capable of reactivating the latter. Probably, a more rigorous classification should be between "recent" and "old" landslide mass, or something similar.

4. Lines 166-168: related to the previous point, the difference between the ranges of electrical resistivity values corresponding to the active landslide material and those of the inactive and old earth-flow seems to be practically negligible. The authors need to provide a more detailed explanation on this point.

5. Line 204: "underlying" should be replaced by "overlying"; the active earth-flow is above the old landslide body.

Some conclusions proposed by the authors are questionable, as for example:

6. Lines 277-279: the efficiency of a drainage intervention cannot be measured by means of the water content measurements and, therefore, by the resistivity values observed between different areas of the landslide material. A drainage intervention works fine only if it is capable of reducing the pore water pressures (or piezometric heads) within the landslide mass, which need to be measured effectively to this purpose.

7. Lines 281-284: what does the sentence "the lithotypes outcropping on the slope, mainly sands and clays, represent the predisposing factor for landsliding" exactly mean? Also, what does the sentence "the increase of water content in the subsoil, due to the occurrence of intense rainfall events, can be considered the triggering factor" mean? Both the previous sentences are too generic and need to be clarified in a more rigorous way.

---

## Referee Comment (RC3) · Anonymous Referee #3 · 2 Apr 2016

General comments:

The manuscripts presents the results of an ERT survey performed on the Montaguto earth-flow, located in the southern Apennines (Campania Region, southern Italy). 11 profiles ERT with a maximal investigation depth of ∼40 meters were performed with the goal to reconstruct the geometry of the landslide body.

The paper is well written, the abstract clear and precise, the figures are well documented and clear. On the other side, the treatment of data can be improved, e.g., a 3D modelling could be included, and the robustness of the inversion should be discussed. How sure are the estimations of the interface between the "old flow" and the "actual flow"? How the location of this interface will change depending on iterations and on the

inversion method (robust or least-square) ? The discussion of results is too general and rather speculative, the conclusions are too strong.

Specific comments:

1. The survey consists of 11 parralel profiles showing coherent features. In my opinion, it would be very helpful to try a 3D inversion of this data set.

2. Figures 2-3: Please give the borehole information in the same terms as ERT profile units (FFa1, 2 etc)?

3. The major remark concerns lines 162-164. The authors "used data from literature, geological surveys and exploratory boreholes to calibrate the ERT and to directly correlate electrical resistivity values with the lithostratigraphic characteristics. " It is not enough to give a reference in such a major point, please detail this translation of the ERT features to the lithostratigraphic characteristics which is a most important point in the paper. Are there some laboratory measurements available on the samples or are the resistivity? Or is the calibration done using outcrops or borehole data? How do you find and how sure you are that 6-12 Ohmm structure corresponds to the activated earth flow (e.g., L228-230) ?

4. L231-234: Observed resistivity values and water content. It lacks a reference here (e.g. Waxman & Smith, 1968). The authors suppose that the higher conductivity of the underlying unit is related to a higher water content and refer to the independent piezometric data coming from boreholes but do not show these data. It would be interesting to show the data on water content and to check your assumption using the formalism given in Waxman & Smith, 1968 (or equations 6-7 in Rinaldi et al, 2010).
* * *

---

## Author Comment (AC1) · 6 Jun 2016

Reviewer #1:

The authors greatly thank the Anonymous reviewer for his comment. Replies to the questions from the reviewer are as follows:

1) Pag.3 line 49. I think that translation of Puglia Region in "Apulia" Region is not completely correct when, in the same phrase, Campania is written in Italian language. Answer: Thanks for the suggestion. We changed "Apulia" with "Puglia".

2) Pag.3 Line52/53 I think that for a completely explanation of the studies in the Montaguto Earthflow, is necessary to insert a citation of the complex monitoring activities

according with (Giordan et Al. 2013, Lollino et Al. 2014) Answer: Thanks for the suggestion. We added the suggested citation.

3) Pag.3 Line59/60 According with the plane of the ERT surveys (fig.2), I think that is more correct: "...was focused on the upper portion of the landslide channel area" because the upper part of the landslide body (from about 750 to 900 m asl) was not covered by ERT surveys Answer: Thanks for the kind advice. We replaced "upper part of the landslide body" with "upper portion of the landslide channel area".

4) Figure 3 On the ERT 3 picture, we observe the S4 borehole but in the legend is not present. The Piezo 1 (P1) is instead present. Answer: Thanks for the kind advice. We added the S4 borehole in the legend.

5) Figure 4 I think that 6 ERT sections in one A4 page are too many. I think that the correct number can be 3 or 4 (as in Figure 3). Answer: Thanks for the suggestion. We changed Figure 4, showing the 11 ERT sections in three different figures.

6) Figure 3 and 4 To increase the comprehension of the drainage channel effects, the pictures are too small. My suggestion: create a zoomed box centered on the drainage channel. The S4,S6,S7 piezometer explain the water table or pore water pressure? The difference between W.T. and P.W.P. is very important on this type of terrains Answer: Thanks for the kind advice. Having changed Figure 4, now the pictures are greater and clearer. S4, S6 and S7 (as well as S8) are boreholes where the water table level was measured. P1 and P2 are piezometers and, as shown by Lollino et al., 2014, the water pressure was measured (see figures and tables in Lollino et al., 2014). We corrected the legend of both Figures (3 and 4). Moreover, the text in the Results section was modified accordingly.

7) Pag.9 - Conclusions - Line 251/252 Regarding the effectiveness of the drainage system, I'm not completely sure that with this surveys is possible to make this important and critical information. I think that the your data explain the first effects of the drainage systems but just with many multi-temporal surveys is possible to explain

the effectiveness of the complex drainage systems installed in this terrain. Therefore, my suggestion to modify the phrase. Answer: Thanks for highlighting this subtle but important difference: the sentence was changed according to the reviewer suggestion as following: "From a geophysical point of view, considering that the material included between the drainage channels is characterized by moderate resistivity values (6 < rho < 12 âĎęm) respect to the more conductive surrounding material, it is possible to hypothesize that the complex drainage system installed on the slope is effective in continuously draining and drying the subsoil. In any case, only a multi-temporal survey (e.g. by using time-lapse ERT and in situ pore pressure measurements) could verify the effectiveness of this intervention."

Please also note the supplement to this comment:
http://www.nat-hazards-earth-syst-sci-discuss.net/nhess-2016-28/nhess-2016-28-AC1-supplement.pdf

―――――――――――――――

---

## Author Comment (AC2) · 6 Jun 2016

Reviewer #2:

The authors greatly thank the Anonymous reviewer for his comment. Replies to the questions from the reviewer are as follows:

1) Lines 46-47: the sentence seems relatively questionable, since it is quite unrealistic that the slope failure started in the mid spring of 2006. Based on the historical data, the lower portion of the earth-flow mass obstructed the SS90 National Road on April 2006 and this means that the earthflow was activated some months before, with a run-out of several hundred of meters. In 2010 there was a strong reactivation of the landslide

[Figure]

mass already accumulated at the toe of the slope in the previous years. Answer: Thanks for the kind advice. According to the reviewer comment, the sentence was modified in: "On 26 April 2006 a large remobilization of earth-flow, with an estimated volume of 6 x 106 m3, covered the SS90 National Road (Guerriero et al. 2013)"

2) Line 97: a maximum value of earth-flow width equal to 420 m seems too high. Probably, the authors refer also to a portion of the landslide mass that invaded the valley located at the right-hand side of the proper valley occupied by the landslide mass, but that portion should not be considered as a real part of the main earth-flow body. Therefore, a more precise value of the landslide width should be about 150-200 m. Answer: We based our statement on Ventura et al. (2011) where a width range between 45 and 420 m was reported. Further, for the 2010 movement Giordan et al. (2013) reported an average (?) width equal to 420 m.

3) Throughout the manuscript the authors distinguish between the active landslide material and the underlying inactive one, only based on the ERT data interpretation. This seems too strong, since there is not a clear instrumental evidence of what is the active part of the landslide. Moreover, for this type of landslides it is quite frequent that recently-activated landslide surges travelling above pre-existing landslide masses that rest along the landslide channel are capable of reactivating the latter. Probably, a more rigorous classification should be between "recent" and "old" landslide mass, or something similar. Answer: Thanks for the suggestion. We decided to distinguish the landslide material as 'active' and 'inactive' according to the terminology used by Guerriero et al. (2014). The manuscript text and the figures have been modified accordingly.

4) Lines 166-168: related to the previous point, the difference between the ranges of electrical resistivity values corresponding to the active landslide material and those of the inactive and old earth-flow seems to be practically negligible. The authors need to provide a more detailed explanation on this point. Answer: Thanks for the kind advice. To calibrate the ERT and to directly correlate electrical resistivity values with the lithostratigraphic characteristics we used data from literature (e.g. in Giocoli et al., 2008
and Mucciarelli et al., 2009 resistivity measurements on the Faeto Flysch were carried out), geological surveys, exploratory boreholes, direct and indirect surveys (e.g. static cone-penetration tests and shallow-seismic profiles from Guerriero et al., 2014) and direct resistivity measurements on outcrops. The manuscript text has been modified accordingly.

5) Line 204: "underlying" should be replaced by "overlying"; the active earth-flow is above the old landslide body. Answer: Thanks for the kind advice. We replaced "underlying" with "overlying".

Some conclusions proposed by the authors are questionable, as for example: 6) Lines 277-279: the efficiency of a drainage intervention cannot be measured by means of the water content measurements and, therefore, by the resistivity values observed between different areas of the landslide material. A drainage intervention works fine only if it is capable of reducing the pore water pressures (or piezometric heads) within the landslide mass, which need to be measured effectively to this purpose. Answer: Thanks for highlighting this subtle but important difference: the sentence was changed according to the reviewer suggestion as following: "From a geophysical point of view, considering that the material included between the drainage channels is characterized by moderate resistivity values ($6 < rho < 12$ âĎęm) respect to the more conductive surrounding material, it is possible to hypothesize that the complex drainage system installed on the slope is effective in continuously draining and drying the subsoil. In any case, only a multi-temporal survey (e.g. by using time-lapse ERT and in situ pore pressure measurements) could verify the effectiveness of this intervention."

7) Lines 281-284: what does the sentence "the lithotypes outcropping on the slope, mainly sands and clays, represent the predisposing factor for landsliding" exactly mean? Also, what does the sentence "the increase of water content in the subsoil, due to the occurrence of intense rainfall events, can be considered the triggering factor" mean? Both the previous sentences are too generic and need to be clarified in a more rigorous way. Answer: According to the reviewer and taking into account the

results obtained by the other authors (Guerriero et al., 2014; Lollino et al., 2014), we modified the conclusions of our paper by removing generic sentences and focusing the attention on the information coming only from geophysical results.

Please also note the supplement to this comment:
http://www.nat-hazards-earth-syst-sci-discuss.net/nhess-2016-28/nhess-2016-28-AC2-supplement.pdf

---

## Author Comment (AC3) · 6 Jun 2016

Reviewer #3:

The authors greatly thank the Anonymous reviewer for his comment. Replies to the questions from the reviewer are as follows:

General comments:

The manuscripts presents the results of an ERT survey performed on the Montaguto earth-flow, located in the southern Apennines (Campania Region, southern Italy). 11 profiles ERT with a maximal investigation depth of 40 meters were performed with the goal to reconstruct the geometry of the landslide body. The paper is well written, the

abstract clear and precise, the figures are well documented and clear. On the other side, the treatment of data can be improved, e.g., a 3D modelling could be included, and the robustness of the inversion should be discussed. How sure are the estimations of the interface between the "old flow" and the "actual flow"? How the location of this interface will change depending on iterations and on the inversion method (robust or least-square) ? The discussion of results is too general and rather speculative, the conclusions are too strong. Answer: Thanks for highlighting this subtle but important issue. We carried out once again the inversion of all 2D data set. We used both least-square and robust inversion constrain, preferring the former because, in our case, subsurface resistivity changes in a smooth manner. Furthermore, we checked also the resistivity models for different iteration numbers. In some cases (e.g. ERT1, ERT2, ERT5 and ERT10) we replaced the resistivity model, due to a slightly improvement of the subsurface resistivity patterns. In any case, we note that the interface between the "old flow" and the "actual flow" does not change significantly. Finally, we modified the conclusions of our paper by removing generic sentences and focusing the attention on the information coming only from geophysical results.

Specific comments:

1) The survey consists of 11 parallel profiles showing coherent features. In my opinion, it would be very helpful to try a 3D inversion of this data set. Answer: Thanks for the kind advice. We tried a 3D inversion of the whole resistivity data set, but the 3D model did not produce realistic resistivity patterns of the underground since the distance between the ERT was much more greater than both the distance between the electrodes and the 2D electrical resistivity model cells. Moreover, the complex topography of the area (i.e. very steep towards North-East section) was also an issue. We decided not to show the result in this paper due to its scarce information content and the presence of artifacts in the 3-D inversion model caused in the use of parallel 2-D lines.

2. Figures 2-3: Please give the borehole information in the same terms as ERT profile

units (FFa1, 2 etc)? Answer: Thanks for the suggestion. We modified and simplified borehole information.

3. The major remark concerns lines 162-164. The authors "used data from literature, geological surveys and exploratory boreholes to calibrate the ERT and to directly correlate electrical resistivity values with the lithostratigraphic characteristics". It is not enough to give a reference in such a major point, please detail this translation of the ERT features to the lithostratigraphic characteristics which is a most important point in the paper. Are there some laboratory measurements available on the samples or are the resistivity? Or is the calibration done using outcrops or borehole data? How do you find and how sure you are that 6-12 Ohm structure corresponds to the activated earth flow (e.g., L228-230)? Answer: Thanks for the suggestion. To calibrate the ERT and to directly correlate electrical resistivity values with the lithostratigraphic characteristics we used data from literature (e.g. in Giocoli et al., 2008 and Mucciarelli et al., 2009 resistivity measurements on the Faeto Flysch were carried out), geological surveys, exploratory boreholes, direct and indirect surveys (e.g. static cone-penetration tests and shallow-seismic profiles from Guerriero et al., 2014) and direct resistivity measurements on outcrops. The manuscript text has been modified accordingly.

4. L231-234: Observed resistivity values and water content. It lacks a reference here (e.g. Waxman & Smith, 1968). The authors suppose that the higher conductivity of the underlying unit is related to a higher water content and refer to the independent piezometric data coming from boreholes but do not show these data. It would be interesting to show the data on water content and to check your assumption using the formalism given in Waxman & Smith, 1968 (or equations 6-7 in Rinaldi et al, 2010). Answer: Thanks for the suggestion. We tried to apply the formalism given in Waxman & Smith (1968) but, unfortunately, we don't have all the required parameters. We have a measurement of porosity for a single depth (at S6, S7 and S8 boreholes) thus we should assume the same porosity throughout the investigated section. Such an assumption would probably smooth out the spatial variations of the subsoil saturation

that we would like to highlight instead. Moreover, data on the type of clays and on their cation-exchange capacity for the estimation of the surface conduction are not available. Our assumption is based on the resistivity values and, as also specified in the text, is supported by the water table levels measured and indicated in the S8, S7 and S6 boreholes and by the pore water pressures recorded at the P1 and P2 piezometers (see Fig.9 in Lollino et al., 2014).

Please also note the supplement to this comment:
http://www.nat-hazards-earth-syst-sci-discuss.net/nhess-2016-28/nhess-2016-28-AC3-supplement.pdf